# Learning English tenses from Sentential Input: A Neural Network Approach

## Abstract

Children are able to productively use and understand tense information, such as the lexical verbs, auxiliaries and copula and tense morphemes. How children acquire the tense information remains unclear. One controversy is whether linguistic input alone is sufficient enough for the children to learn these tense information, or whether the children extract these information using abstract syntactic knowledge and/or multimodal cognition. This study uses transformer models to understand the process of tense acquisition from the sentential input. We train transformer models on English tense classification tasks with sentences in child directed speech as the input. When the transformer models successfully learn the tense, we find that 1) the models are sensitive to auxiliary verbs (e.g. *was*, *do*) but not phrases (e.g. *is going to*), 2) the past tense *-ed* form facilitates classification, and 3) temporal adverbs have limited impact in tense classification.

## 1 Introduction

Children are able to understand and produce verb forms with tense/aspect at an early age. In comprehension, 2-year-old children are able to use auxiliaries (*will/did*, *is/was*) and copula (*is/was*) to distinguish the past and present (or future) version of a scene, e.g. 'Show me the crayons that *is/was* rolling' (Wagner, 2001; Valian, 2006). Children around 2.5 years are able to use the tense/aspect morphology on nonce verbs to choose between the present ongoing and completed events, e.g. 'She's *kradding* it.', 'She *geeded* it.' (Wagner et al., 2009). In production, English speaking children start to use past tense verbs with high accuracy at around age 2 years (e.g. Brown, 1973). Children around age 3 years begin to produce overgeneralization errors on regular English verbs, e.g. *\*holded, \*feeled* (Marcus et al., 1992; Maratsos, 2000), as well as applying the '*-ed*' form to the nonce verbs, e.g. 'It *pudded* my knee.' (Akhtar and Tomasello, 1997),

suggesting that they have the knowledge of the past tense '*-ed*' morpheme.

Although much is known regarding the emergence of these forms, the literature remains limited in its exploration of how these forms emerge, in particular how the verbal morphology emerged from sentential input. In real life acquisition, children almost never hear the isolated 'stem verb - tensed verb' pairs such as '*help - helped*' where they could easily extract the tensed morpheme. Instead, they hear these forms in sentences in different contexts, e.g. 'If you *ask* nicely, I'll give it to you.' and 'I *asked* you to clean it up'. In this scenario, how are the tensed forms constructed from these sentences, and can they be constructed from linguistic input alone without innate abstract syntactic knowledge or/and multimodal cognition? One hypothesis is that children initially rely on the lexical forms of the verbs such as '*did*', '*is*' in their tense acquisition, which motivates the studies like Wagner (2001) and Valian (2006) to test children with auxiliaries and copula verbs. The other hypothesis is that temporal adverbs could facilitate children's understanding of tenses. However, the contribution of the adverbs is difficult to interpret. In children's spontaneous speech, temporal adverbs occur later in development than verb inflections (Smith, 1980). In addition, Wagner (2001) and Valian (2006) found that adverbs had little impact on 2-year-old children's understanding of tenses.

In this study, we propose a neural network approach to understand the process of the emergence of the tensed information from sentential input. We train transformer models on tense classification tasks with parents' sentences as input. When the models successfully classify the tenses, we investigate how the model make these classifications. In particular, we ask 1) whether the models are sensitive to lexical verbs, auxiliaries and copula and phrases (e.g. *went*, *did*, *is*, *are going to*); 2) whether the tense morphemes (e.g. *'-ed'* and third

person agreement '-*s*') facilitate the models' classification; 3) whether the temporal adverbs (e.g. *now*, *yesterday*, *tomorrow*) improve the model's classification.

## 2 Background

### 2.1 Learning English Tenses

In English, there are only two grammatical tenses - present (or non-past) and past. The future time often is expressed via modal verb *will* or phrase *be going to*. Certain lexical verbs can indicate tense (or time) of a sentence, such as '*went*', '*let*' etc. However, since the future time is expressed with the present grammatical tense, many auxiliaries can be ambiguous, e.g. 'I am crying.' vs 'I'm leaving in an hour.' Verb inflections are usually used to distinguish tenses, e.g. *help - helped*. However, many of the most commonly used English verbs are irregular verbs. Their inflections can not reliably be used to distinguish tenses, e.g. *put - put* and *go - went*. In addition, temporal adverbs convey tense information, e.g. *last night*, *all the time*. However, many of the temporal adverbs can be used in more than one tense (or time), e.g. 'Now we'll see.', 'I'm doing it now', and 'Now you broke it.'. Therefore, the linguistic features seem to be not reliable to distinguish tenses, which creates difficulties in understanding tenses from the sentential input.

### 2.2 NLP Approach in Tense Classification

Although tense understanding poses challenges in language acquisition, it is not a difficult task in the field of NLP. Much of the previous work on tense classification has been for the purpose of improving machine translation, abstract meaning representation and text generation. Ye and Zhang (2005) and Ye et al. (2006) explored tense classification of Chinese sentences with machine learning approaches using conditional random fields with a combination of features including verb telicity, verb punctuality and temporal ordering of the events. Ramm et al. (2017) constructed a rule-based model that operates on the dependency parsers for annotating verbs with tense, mood and voice in English, German and French. Myers and Palmer (2019) trained a bidirectional LSTM-CRF model that successfully identifies tenses and aspects of verbs in English and outperforms the rule-based model. In addition to the classification tasks, Logeswaran et al. (2018) trained an encoder-decoder model that is capable of changing the tense of a given sentence, as well as changing the mood, complexity and voice.

These previous work showed that classifying sentence's tense can be easily achieved by NLP models. However, the cognitive implication of these studies are very limited . First, most of the models are trained on a large amount of data[1], which does not truly reflect the reality in children's language acquisition. Second, these studies provided little analysis of the results of the classification, since most of them are using the tense classification as a means to improve other NLP tasks. In our study, we intend to provide a more detailed analysis on the model's classification results in order to understand how the tense is classified given sentential input.

## 3 Data

### 3.1 Corpus Data

We use Adam's data from Brown corpus (Brown, 1973) in the CHILDES database (MacWhinney, 2000) as a case study for our model training. Adam's recording starts at the age of 2;3. He made the first overregularization error at the age of 2;11, 'What dat *\*feeled* like?'. This error implied that Adam had already constructed the past tense *-ed* form. Therefore, we select Adam's parents' sentences (including both mother and father) from age 2;3 to 2;11 as our training dataset. We categorize these sentences into four classes following the previous NLP work: a) No tense: there is no inflected verb in the sentence, e.g. 'not in your mouth', 'sitting in Adam's chair'; b) Present tense: the inflected verb in the sentence is a present tense verb (except for the future time phrase 'is/am/are going to'), e.g. 'where are you going', 'you tell me'; c) Past tense: the inflected verb in the sentence is a past tense verb, e.g. 'did it hurt', 'who fell down'; d) Future time: the sentences include modal verb 'will/won't' and phrase 'be going to', e.g. 'will that fit in here', 'you're going to build the house'. For complex sentences, we divide the sentences into separate clauses and annotate them respectively, e.g. original sentence: 'can't you tell us what happened?' - sentence 1: 'can't you tell us' - present, sentence 2: 'what happened' - past.

---

[1]For example, (Myers and Palmer, 2019)'s bi-LSTM mdoel was trained on PropBank corpus with 112,570 verb tokens.

| Class | Sentence Count | Mean Sentence Length | Tensed Verb types | Percentage of sentences with a time adverb |
|---|---|---|---|---|
| No Tense | 2039 (26.1%) | 2 | | |
| Present | 4500 (57.6%) | 4.9 | 228 | 2.00% |
| Past | 1068 (13.7%) | 5.3 | 96 | 3.09% |
| Future time | 203 (2.3%) | 6.8 | 13 | 7.88% |

Table 1: The summary of counts, length, verb types and adverb percentage of different tense classes

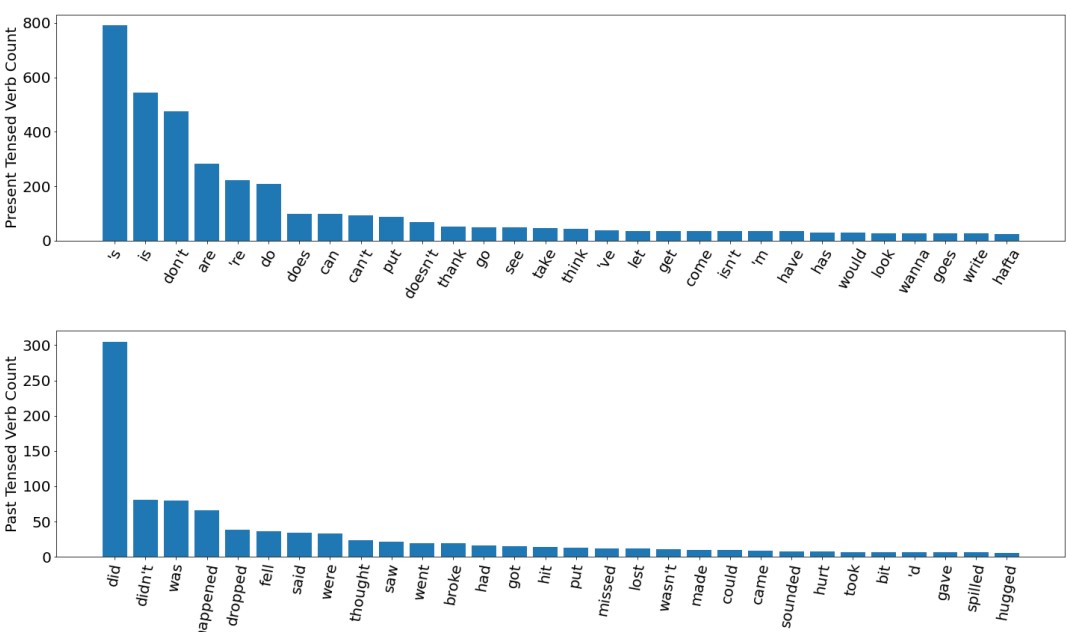

Figure 1: The frequency distribution of most commonly 30 tensed verbs of present tense and past tense

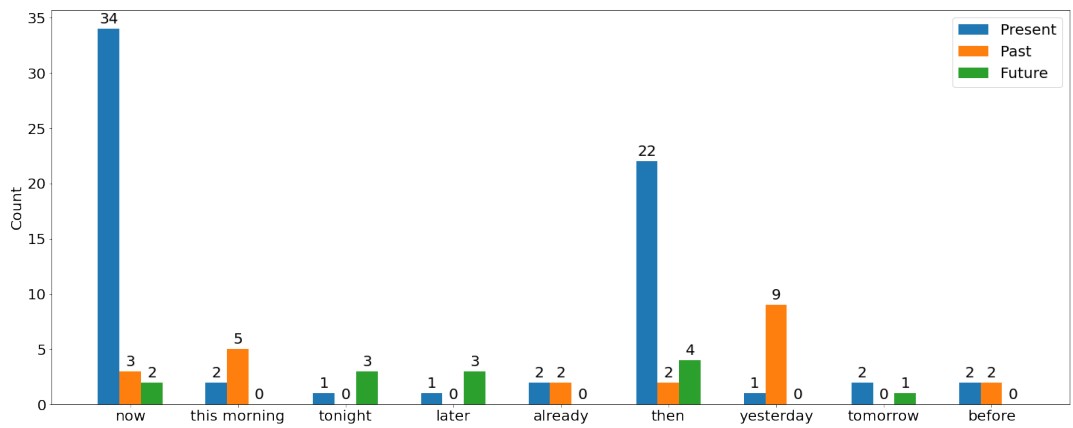

Figure 2: Frequency Distribution of temporal adverbs in different tensed sentences

### 3.2 Data Description

There are 7810 sentences in Adam's parents' input, with 6229 unique sentences. The summary of the descriptions of the sentences is shown in Table 1. The present tense class has the most number of sentences (4500), followed by the no tense class (2039), the past tense class (1068) and the future time class (203). Most of the sentences are very short, with an average length of 4.3 words and a max length of 21 words. The No Tense class sentences have the shortest length, with a mean length of 2. The Future class sentences have the longest length, with a mean length of 6.8.

**Tensed Verb Types:** In order to see if different tense classes consist of different verbs, we count

the types of tensed verbs in each class. Most of the tensed verbs in the parents' sentences are the auxiliaries *be* and *do*. For the present tense sentences, the most commonly used verbs are *'s/is* and *don't*. For the past tense sentences, the most commonly used verbs are *did/didn't* and *was*. The frequency distribution of the most 30 common tensed verbs for each tense class is shown in Figure 1. The tensed present verbs and the tensed past tensed verbs are distinct from each other. It's possible to differentiate the tense of the sentences based on these lexical verbs. However, it is unclear how the past tense '*-ed*' can be derived. For the most commonly 30 verbs, there are several present-past pairs such as 'do/does - did', 'put - put', 'take - took', 'think - thought', but most of them are irregular verbs that don't contain the '*-ed*'.

**Temporal Adverbs:** We also count the temporal adverbs in the tensed sentences. There are total 29 types of temporal adverbs, and the most frequent ones are *now*, *then* and *yesterday*. Parents' tensed sentences rarely contain a temporal adverb. Only 2% of the present tensed sentences have a temporal adverb, and 3% of the past tensed sentences have a temporal adverb and 8% of the future time sentences have a temporal adverb. In addition, 9 of the adverbs appear in more than one type of tensed sentences. For example, *now* appeared in 34 present tensed sentences (e.g. 'Now who is it'), 3 past tensed sentences (e.g. 'No she gave you a piece now.') and 2 future time sentences (e.g. 'Now I'll wait for you to come in.'). The distribution of these time adverbs in different tensed sentences is show in Figure 2. Given the rarity and the distribution of temporal adverbs, they might not be a very informative feature in tense classification.

## 4 Model

### 4.1 Model Architecture

Transformer models have been shown to be useful for numerous sequence-based tasks, such as machine translation (Vaswani et al., 2017). We expect good performance on classification of tense with transformer models. Since the dataset for our tense classification task is significantly smaller than traditional transformer tasks, we employ a smaller transformer with 2 layers in the encoder (1 attention layer, 1 feed-forward layer) followed by one dense layer for classification. Layer normalization is applied to the output of encoder and the dense layer. Positional embedding layers are used to capture the positional information. We use 4 self-attention heads, with an embedding size of 256 and a hidden size of 128 for the feed-forward layer.

### 4.2 Model Training

We train three models with different RoBERTa-based tokenizers in our experiment. The RoBERTa model (Liu et al., 2019) has the same architecture as BERT model, but uses byte-pair-encoding (BPE) as the tokenizer and uses a different pretraining scheme. The BPE tokenizer is a type of subword-based tokenization that combines word-level and character-level tokenization. The BPE tokenizer extracts the most common pairs of consecutive bytes of data, which makes it possible to tokenize the frequent inflectional morphemes, e.g. 'lowest' - 'low' + 'est</w>'.

We first use the RoBERTa tokenizer in our experiment. However, RoBERTa tokenizer is trained on 30 billion English words with 125 million parameters, which operates in a very different regime than language-learning children. Therefore, we also use BabyBERTa tokenizer to better simulate the children's learning. BabyBERTa is a RoBERTa-based model trained on 5 million words of parents' input in the CHILDES dataset (Huebner et al., 2021). For our experiment, we suspect that the BabyBERTa tokenizer might also be too big of a model since it includeds all the parents' input between the age of 1 to 6. In order to represent the input of the children by the time they start to understand tenses, we also train a 2y/o-BabyBERTa with the same parameters as BabyBERTa only including parents' input before 2 years of age. The summary of the parameters of the classification models with different tokenizers is shown in Table 2.

| Tokenizer | Training Words | Vocab Size | Params |
|---|---|---|---|
| RoBERTa | 30B | 50,266 | 13M |
| BabyBERTa | 5M | 8,193 | 2.4M |
| 2y/o-BabyBERTa | 1.8M | 501 | 0.4M |

Table 2: Summary of classification model's parameters with different tokenizers

The train-dev-test-split ratio is 80-10-10. The training data include 6248 sentences and validated on 781 sentences. The training was done using Adadelta optimization with batch size of 16. We train 50 epochs for each model.

## 5 Results

### 5.1 Classification Accuracy

We first evaluate the model's accuracy in tense classification on the test-split dataset. In general, all three models achieved good performance on classification tasks with the accuracy over 90%. The overall accuracy and accuracy for each class is summarized in Table 3. The confusion matrix for three models are shown in Figure 3 - 5 in Appendix.

### 5.2 Error Analysis

There are 15 sentences in the testing dataset that all three models made errors. These sentences are listed in Table 8 in the Appendix. 2 of these sentences belong to No Tense class, and all the models predicted them to be the present tense. Both sentences contain the possessive ''-s'. The models might mistake the possessive '-s' as the copula '-s'. 1 sentence is in the Future Class with '*is going to*' phrase, and all three models classified it as a present tense sentence. The model might only focus on the auxiliary '*is*' and ignored '*going to*'. There are 4 sentences in the Past Tense class, and all three models labeled them as the present class. 3 of these sentences contain a regular past tense verb ('*ticked*', '*popped*' and '*touched*') and 1 sentence contains the irregular verb '*put*'. It is reasonable for the models to classify the sentence with '*put*' as a present tense sentence since the past tense form is the same as the present form. The failure to classify the three regular past tense might indicate that the models do not have the robust knowledge of the past tense form '*-ed*'.

There are 8 present tensed sentences that all three models mis-classified as other tense classes. 5 of these sentences were labeled as no tense by the models. 4 of these sentences don't have a subject (e.g. 'fix kitty'). The model might be sensitive to the argument structure in tense classification. 3 of the present sentences were labeled as the past tense. 2 of these sentences are complex sentences that the relative clause contains a past tensed verb (e.g. 'You know where it went'). 1 sentence contains the ambiguous verb '*hurt*' that was labeled as past tense too.

The preliminary analysis of the models' classification errors showed that the models might be sensitive to lexical verbs, auxiliaries and copula, but not necessarily the future time phrase *be going to*. In addition, the models also showed not so robust knowledge of the tense morphemes.

## 6 Testing on Nonce Verb Sentences

### 6.1 Nonce Verb Sentence Dataset

We create a tensed sentence dataset with nonce verbs to better evaluate the models' classification results. We select 54 nonce verbs in Albright and Hayes (2003). Each of the verb was carefully constructed to have some phonological similarity of existing English verbs and has a regular past tense form and an irregular one, e.g. '*bize - bized/boze*'. We use these verbs to create sentences with present tense, past tense and future time. Examples of these sentences are listed in Table 4. The present tense has 2 types: the first person present tensed verb and the third person present verb with '*-s*' agreement. 4 types of past tense sentences were created, including the sentences with the regular past tense verbs with '*-ed*', the irregular verbs, the past tense auxiliary '*did*' and the past progressive sentences with the auxiliary '*was*'. There are 2 types of the future time sentences: with modal verb '*will*' and with the phrase '*is going to*'.

The dataset aims to test two hypotheses. First is that whether the models are sensitive to the lexical verbs and phrases in classification. If it is true, we expect to see high accuracy on the classes with verbs '*did*', '*was*' and '*will*', and low accuracy on present verb class and irregular past tense class. The second hypothesis is about whether the models rely on verb morphemes to classify tense. If this is true, we expect to see better accuracy in the regular past tense class comparing to the irregular past tense class, and better accuracy for the present tense class with '*-s*' than the regular present tense class.

### 6.2 Results

The overall accuracy for the nonce sentence is worse than the testing dataset, since the accuracy is only around 50% for the three models. The accuracy for each type of the tense is summarized in Table 5.

**Hypothesis 1**: Are models sensitive to the lexical verbs and phrases? All three models achieved almost perfect accuracy on the tense classes with '*did*', '*will*'. For the auxiliary *was*, the model with RoBERTa and BabyBERTa tokenizer achieved almost perfect accuracy. The model with 2yo-BabyBERTa tokenizer had an accuracy of 0.15 since it mislabelled most of the sentences in this class as the future time. In addition, the future class *is going to* were all mislabelled as present tense in

| Model | No Tense | Present | Past | Future Time | Overall |
|---|---|---|---|---|---|
| RoBERTa | 0.92 | 0.93 | 0.91 | 0.83 | 0.92 |
| BabyBERTa | 0.92 | 0.94 | 0.90 | 0.83 | 0.93 |
| 2yo-BabyBERTa | 0.91 | 0.92 | 0.84 | 0.72 | 0.90 |

Table 3: The classification accuracy of different model of the testing dataset

| Sentence | Type |
|---|---|
| I *bize* the door. | present |
| She *bizes* the door. | present -*s* |
| I *bized* the door. | past - reg |
| I *boze* the door. | past - irr |
| She did *bize* the door. | past - did |
| She was going to *bize* the door. | past - was |
| She will *bize* the door. | future - will |
| She is going to *bize* the door. | future - phrase |

Table 4: Example sentences of the different tensed sentences

| Types | RoBERTa | Baby | 2yo |
|---|---|---|---|
| Present | 0.65 | 0.19 | 0.52 |
| Present -*s* | 0.74 | 0.31 | 0.35 |
| Past -*ed* | 0.67 | 0.59 | 0.80 |
| Past *did* | 1.00 | 1.00 | 1.00 |
| Past -irr | 0.06 | 0.06 | 0.04 |
| Past -*was* | 0.98 | 0.98 | 0.15 |
| Future -*will* | 0.98 | 1.00 | 1.00 |
| Future - *is going to* | 0.00 | 0.00 | 0.00 |
| Overall | 0.59 | 0.52 | 0.48 |

Baby = BabyBERTa,
2yo = 2yo-BabyBERTa

Table 5: Summary of accuracy of different tense types with three models

all three models. This suggest that the model probably did not process *be going to* as a unit to indicate future.

**Hypothesis 2**: Do models rely on verb morphemes to classify tense? For past tense classes, we compare the accuracy of past -*ed* class, the past -irr class. All three models' accuracy on past -*ed* classes are higher than the irregular past tense classes. This suggests that the model are sensitive to the past tense '-*ed*' morpheme.

In addition, we found that three regular past tense verbs that all three models mislabelled as the present tense: '*gared*', '*preaked*' and '*scoiled*'. To test if we can get the model to predict the correct past tense label, we first add temporal adverbs such

as '*yesterday*', '*last night*' in the sentences. We use the Language Interpretability Tools (LIT) (Tenney et al., 2020) to test sentences with temporal adverbs and output the probability score for the past tense class. The summary of past tense class the probability score is shown in Table 6. We expect the past adverbs '*yesterday*' and '*last night*' would increase the past tense probability score in these sentences and eventually change the classification results to the past tense class. The model with BabyBERTa tokenizer was not affected by the temporal adverbs, that the probability score changes were minimum. Adding *'yesterday'* made the model with 2y/o-BabyBERTa toeknizer to change the classification results, but not the model with the RoBERTa tokenizer. Instead, '*last night*' was able to change the model with RoBERTa's classification result but not the model with 2yo-BabyBERTa tokenizer. This result suggests that temporal adverbs have limited affects on models.

Furthermore, we add an extra past tense morpheme '-*ed*' to the verbs to see if the past tense probability score would increase, since the model showed sensitivity to the '-*ed*' morpheme. The double '-*ed*' improves the past tense class probability scores for all three models that most of them correctly classify it as a past tense sentence.

For the present tense classes, we compare the accuracy of the first person present verb and the present verb with -*s*. For the model with RoBERTa and the BabyBERTa tokenizer, the accuracy for the present -*s* class is higher than the regular present class, suggesting that the third person agreement morpheme -*s* facilitates to model to classify present tenses. However, the model with 2y/o-BabyBERTa tokenizer had worse accuracy in present -*s* class than the regular present class, suggesting that for this model the present morpheme '-*s*' did not help classification.

**Temporal Adverbs in Future time:** Since all the future sentences with *is going to* have been mislabelled by all models, we add future adverbs *tomorrow* and *next week* to see if the probability scores for the future class would increase. The re-

|  | Probability Score for Past Tense Class | | |
|---|---|---|---|
| Sentence | Ro | Baby | 2yo |
| I gared the door | 0.176 | 0.193 | 0.175 |
| ~yesterday | 0.176 | 0.211 | 0.440* |
| ~last night | 0.475* | 0.183 | 0.177 |
| I *gareded* the door | 0.475* | 0.475* | 0.475* |
| I preaked the door | 0.227 | 0.175 | 0.175 |
| ~yesterday | 0.232 | 0.175 | 0.175 |
| ~last night | 0.475* | 0.175 | 0.175 |
| I *preakeded* the door | 0.475* | 0.462* | 0.431* |
| I scoiled the door | 0.175 | 0.175 | 0.193 |
| ~yesterday | 0.176 | 0.175 | 0.424* |
| ~last night | 0.455* | 0.175 | 0.175 |
| I *scoileded* the door | 0.431 | 0.475* | 0.475* |

Ro = RoBERTa, Baby = BabyBERTa

2yo = 2y/o-BabyBERTa

* indicates the model successfully labels it as past tense

Table 6: The probability of past tense class for different sentences with temporal adverbs and double *-ed*

sults are summarized in Table 7. The future adverbs almost had no impact on the future class probability scores for the models.

|  | Ro | Baby | 2yo |
|---|---|---|---|
| She is going to nold | 0.175 | 0.175 | 0.184 |
| ~tomorrow | 0.175 | 0.175 | 0.175 |
| ~next week | 0.175 | 0.175 | 0.179 |

Table 7: The probability score of future class with future adverbs

## 7 Conclusion

In this study, we train transformer models with different tokenizers to classify the tense of parents' input sentences. With a small amount of data, the models successfully classify the tenses, with an overall accuracy of around 90%. By analyzing the errors on the classification and testing the sentences with nonce verbs, we find that the models are sensitive to lexical verbs, auxiliaries and copula, but not phrases like *be going to*. This result suggests that the models are likely to rely on single words in classification, but not the phrases. In addition, we also find that the tense morphemes facilitate the models' classification, especially the past tense *-ed* form. This result suggested that the morpheme-level information can be extracted from sentential input

with subword tokenizers. Moreover, the temporal adverbs have little impact on models' classification, which is similar to the findings in children's tense understanding.

In addition, this study also shows that linguistic input alone might be sufficient enough to extract tense information, since our models were not given other information. Although the transformer models do not represent children's acquisition mechanisms, we hope this study could provide some insight in understanding the acquisition process of tense.

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

## A  Appendix

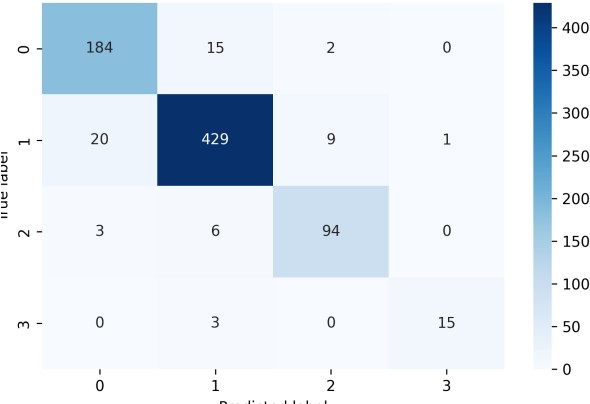

Figure 3: Confusion Matrix of the model with **RoBERTa** tokenizer on test-split

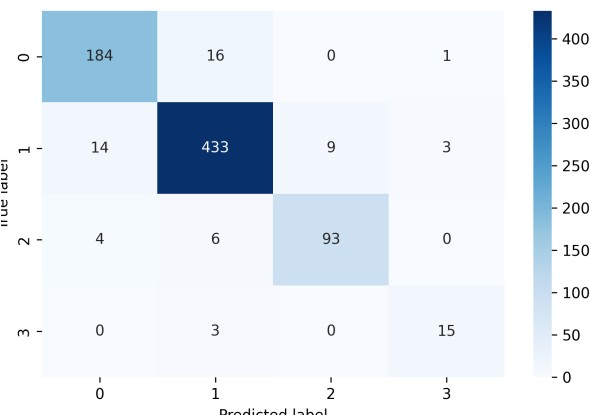

Figure 4: Confusion Matrix of the model with **Baby-BERTa** tokenizer on test-split

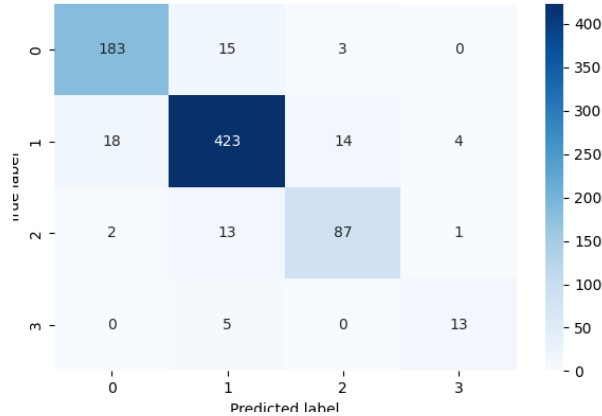

Figure 5: Confusion Matrix of the model with **2y/o-BabyBERTa** tokenizer on test-split

| Label | Sentence | Predict |
|---|---|---|
| 0 | cowboy's grass | 1 |
|   | oh Timmy's boots | 1 |
| 1 | fix kitty | 0 |
|   | little boy play with David | 0 |
|   | read bunny | 0 |
|   | just touch that | 0 |
|   | squeeze your own | 0 |
|   | you hurt the floor | 2 |
|   | you know where it went | 2 |
|   | oh that's who we thought it was | 2 |
| 2 | you put them in your bank | 1 |
|   | what tickled | 1 |
|   | his fingers popped either | 1 |
|   | you touched Cromer | 1 |
| 3 | Mommy is going to stay tonight | 1 |

0 = No Tense, 1 = Present, 2 = Past,
3 = Future

Table 8: 15 sentences in the test split that all models predicted wrong

