# OpenReview forum: "Learning English tenses from Sentential Input: A Neural Network Approach"
_aclweb.org/ACL/2022/Workshop/CMCL — Submitted to CMCL 2022_

### Official Review · Reviewer_UzZu · 2022-03-23
**Some good ideas, but not sure about the point**

**Rating:** 4
**Confidence:** 4

**Review:**

This paper asks whether linguistic input alone (i.e., without situational context) is sufficient for acquiring English tense. They hypothesize that it is, and test with three related transformer models, RoBERTa, which uses BPE to capture sub-word information, BabyBERTa, which is trained on English CHILDES, and a 2yo-BabyBERTa trained on a fraction of CHILDES. They conclude that there results support the hypothesis.

I like that the authors trained on CHILDES and used BabyBERTa. The recognize that RoBERTa, which trained on orders of magnitude more data than a learner would receive by age 2-3 may overachieve for that reason. 2yo-BabyBERTa seems unecessary to me. BabyBERTa was trained on 5M words, but that's just a large fraction of what an individual learner would hear. It seems unnecessary to restrict it further.

I'm worried about the posibility of test-on-train here. Isn't 5M words pretty much all of English CHILDES? Can the authors be sure that BabyBERTa (or even RoBERTa? I don't know) weren't trained on the subset of CHILDES that they're testing on?

The authors report results on a held out (but see above) test set and a set of sentences with nonce verbs. They find very good (.90-.92 accuracy) on the held out set, and middling performance (0.48-0.59) on the nonce set.

I don't believe the author's conclusion is motivated for two reasons:

First, they argue based on the results on the held-out test set, that since the transformers learn to distinguish tenses well, that this shows that language input alone is sufficient. However, they could've come to the opposite conclusion based on the nonce study. A contrarian could easily conclude language alone is insufficient.

Second, and I think more profoundly, is that there's no particular reason to believe that transformers are good models of human learning. The authors rightly acknowledge this: "Although the transformer models do not represent children’s acquisition mechanisms, we hope this study could provide some insight in understanding the acquisition process of tense."

So we can't conclude anything about acquisition from this study, other than, perhaps, that training on a big chunk of CHILDES works almost as well as training on 30B tokens of miscellaneous text for this purpose.

---

### Official Review · Reviewer_71ka · 2022-03-23
**English tense information is mostly carried by auxiliaries in child-directed speech**

**Rating:** 6
**Confidence:** 3

**Review:**

This paper describes a simulation study in learning to predict tense from CDS sentences from the Adam corpus. The model uses auxiliaries and the past suffix -ed to detect tense, but does not learn morphologically irregular past tenses nor the periphrastic future "going to". These effects are linked to trends in input frequency as well as analyses from the human learning literature.

I found the paper very interesting in its focus on language acquisition of an important semantic distinction and in linking explicitly to work on human learning. I also appreciated the use of nonce verbs and various probing techniques in analyzing the reuslts.

I did have some problems in following the argumentation.

1: Line 63 says "the other hypothesis is that temporal adverbs could facilitate" the learning of tense. But I don't see where this hypothesis comes from. The cited research on the following lines suggests that children don't pay much attention to temporal adverbs. So whose hypothesis is this and why is it worth studying?

2: I am a bit confused about the model setup. My understanding is that the model is a 2-layer transformer without any pretraining (section 4.1) and that only the tokenizers are adopted from pretrained LMs (section 4.2). Table 2 shows the training data sizes of three models, Roberta, BabyBerta and 2y/o BabyBerta and also their number of parameters (which should be irrelevant if no pretraining is used). I think these pretrained models are used only to create the tokenizers, not the model itself. However, I am not perfectly clear on this point, especially because later tables 3 and 5 report performance by "Model" ranging across Roberta, BabyBerta and 2y/o.

3: Following from this, if no pretraining is used, I am not sure why this decision was made. Pretraining followed by fine-tuning is known to learn more robust semantic representations than just learning to classify and would potentially do a better job of extracting the lexical semantics of irregular verbs based on their distribution. Why train a new transformer from scratch instead of using BabyBerta itself as a model and adding a classifier layer on top of it?

4: Although the three tokenizers are compared throughout, there is no clear conclusion about the differences between them; I am not sure in the end why this comparison is important.

---

### Official Review · Reviewer_R9j9 · 2022-03-24
**Interesting work**

**Rating:** 7
**Confidence:** 4

**Review:**

This work uses a transformer model to investigate learning the past tense from child-directed speech. The study focuses on whether the model variants can exploit information from morphology, auxiliaries, or temporal adverbs.

I found the study very interesting and relevant, and a very good fit for this venue.

There are a number of points that could be improved. The nonce test is definitely relevant, but it is based on a very small number of items. I also found the paper could do a better job at situating the results.  It is unclear to me what we learn from comparing the 3 tokenizers. I also missed some reflection on what the results entail for language acquisition.The authors almost apologetically mention that the model is not cognitively plausible, but I think that is not necessarily a downside: these are conclusions from a data-driven model, which are relevant to gather insight on which linguistic information can be exploited. Some discussion on this would make the paper stronger.

---

### Decision · Program_Chairs · 2022-03-29

Reject